# EmbedKGQA: Improving Multi-hop Question Answering over Knowledge Graphs using Knowledge Base Embeddings

## Reproducibility Summary

**Scope of Reproducibility**

Our work consists of four parts: (1) Reproducing results from Saxena et al. [2020] (2) Adding more experiments by replacing the knowledge graph embedding method (3) and exploring the question embedding method using various transformer models (4) Verifying the importance of Relation Matching (RM) module. Based on the code shared by the authors, we have reproduced the results for the EmbedKGQA method. We have not performed relation matching deliberately to validate point-4.

**Methodology**

We have used the code provided by Saxena et al. [2020] with some customization for reproducibility. In addition to making the codebase more modular and easy to navigate, we have made changes to incorporate different transformers in the question embedding module. Question-Answering models were trained from scratch as no pre-trained models were available for our particular dataset. The code for this work is available on GitHub[1].

**Results**

We were able to reproduce the Hits@1 to be within $\pm\mathbf{2.35\%}$ of reported value (in most cases). Anomalies were observed in 3 cases. [1] In MetaQA-KG-Full (3-hop) dataset [2] In WebQSP-KG-50 and [3] WebQSP-KG-Full datasets. From our experiments on the QA model, we have found that a recent transformer architecture, SBERT (Reimers and Gurevych [2019]) produced better accuracy than the original paper. Replacing RoBERTa (Liu et al. [2019]) with SBERT (Reimers and Gurevych [2019]) increased the absolute accuracy by ≈3.4% in half KG case and ≈0.6% in the full KG case. (KG: Knowledge Graph, "≈": Approximately)

**What was easy**

As the code was open-sourced, we didn't have to implement the paper giving us the liberty to customize the codebase to focus on the author's claim validation, perform extended experiments and explore shared as well as new models. In addition to this, pretrained KG embedding models were shared which helped in the reproduction experiment.

**What was difficult**

The lack of a comprehensive documentation alongwith missing comments defining functions/classes/attributes etc., made it laborious to review the code and modify it. In addition to large training times for question answering models, the knowledge graph embeddings also required a significant amount of computing resources.

**Communication with original authors**

We had a couple of virtual meetings with Apoorv Saxena [2], the primary author of EmbedKGQA.

---

[1] `https://github.com/jishnujayakumar/MLRC2020-EmbedKGQA`
[2] `https://apoorvumang.github.io`

# 1 Introduction

Knowledge is the key to question answering task. Knowledge Graph (KG) is a multi-relational graph consisting of entities as nodes and relations among them as typed edges. KGs can accommodate a wide variety of facts, making them one of the potential candidates for intelligent decision-making. Question Answering over KG (KGQA) task aims to answer natural language queries posed over the KG. Multi-hop KGQA is a trending topic and has gained traction from both academia and industry recently. Multi-hop KGQA task involves reasoning over multiple edges of the KG to arrive at the correct answer.

Earlier works on KGs(e.g. Suchanek et al. [2007], Google [2013], Lehmann et al. [2015], Mitchell et al. [2018]) have some element of sparsity, i.e. they do not capture all the facts available in the real world. Recent research on multi-hop KGQA has attempted to reduce this sparsity with the help of relevant external textual resources that are not readily available. On the other side, KG embeddings have emerged as an effective tool to overcome the KG sparsity by predicting missing links in the KG. Although effective, KG embeddings have not been explored for the multi-hop KGQA task. Saxena et al. [2020] fills this gap with the proposed EmbedKGQA method.

This work intends to reproduce and perform an ablation (removing relation matching module) as well as extended study on EmbedKGQA(Saxena et al.[2020]). EmbedKGQA claims to be the first of its kind to use KG embeddings for multi-hop KGQA and improves over other state-of-the-art (SOTA) baselines.

# 2 Scope of reproducibility

According to Saxena et al. [2020], using ComplEx (Trouillon et al. [2016]) KG embeddings significantly improves Hits@1 for multi-hop KGQA task and it has been proved with the help of the results on MetaQA (Zhang et al. [2018]) and WebQSP (Yih et al. [2016]) datasets. This reproducibility work tries to test this claim and conducts experiments as mentioned in table:{2,3} of the original paper. Section 4.1 contains the corresponding results which support the claim with some anomalies.

# 3 Methodology

The authors of the original paper have open-sourced the code along with the data and pre-trained ComplEx KG embedding models. We have used the same codebase (commit:5d8fdbd4) and customized it for our purposes. In addition to this, we have added a comprehensive documentation to make it more interpretable. Moreover, a command-line functionality is also added to easily configure various transformers models in the training workflow.

## 3.1 Model descriptions

As shown in Figure:1, EmbedKGQA has three modules:

1. KG Embedding Module - This module contains a KG embedding model called ComplEx (Trouillon et al. [2016]) to learn embeddings for all entities in the input KG. 4 pretrained models have been shared by the author which contains 2 models for MetaQA-KG-{Full, 50} as well as 2 models WebQSP-KG-{Full,50} dataset. Details about the dataset are mentioned in section:3.2.

2. Question Embedding Module: Given a question q, head entity h and set of answer entities A, this module learns the question embeddings based on the score function defined by the KG embedding method used in 1.

3. Answer Selection Module: This module uses the outputs of module:1 and 2 to select the final answer by scoring the (head entity and question) pair against all possible answers. The strategy is mentioned in section:{4.4, 4.4.1} of the original paper respectively.

## 3.2 Datasets

There are two datasets used in the original paper. MetaQA (Zhang et al. [2018]) and WebQSP (Yih et al. [2016]). Both datasets have two portions. (1) KG data (2) QA data. KG data for both are further divided into two categories. (1) Using the full KG (indicated by suffix KG-Full) and (2) Using only 50% of the facts in the respective KGs (indicated by suffix KG-50). The details of generating custom KG datasets are discussed here[3]. Both datasets are taken from here[4]. Statistics for table:{1, 2} have been taken from Saxena et al. [2020]. For generating question embeddings the

---

[3]`https://github.com/malllabiisc/EmbedKGQA`

[4]`https://drive.google.com/drive/folders/1RlqGBMo45lTmWz9MUPTq-0KcjSd3ujxc` (As of 01/2021)

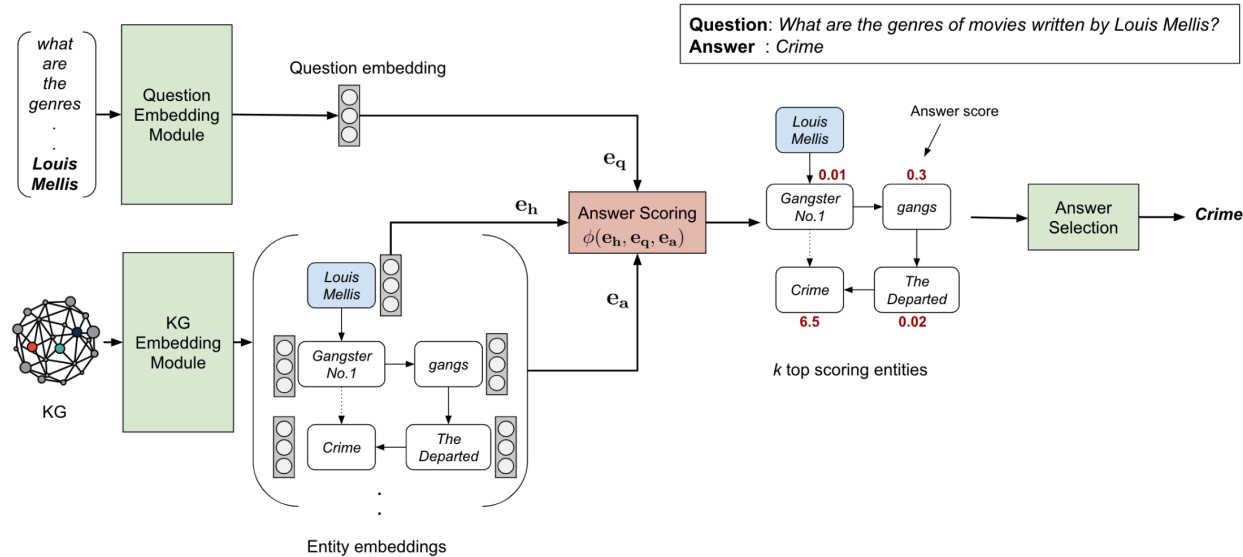

Figure 1: Overview of EmbedKGQA, the proposed method for Multi-hop KGQA.
Image source: Saxena et al. [2020].

| Dataset | Train | Dev | Test |
|---|---|---|---|
| MetaQA 1-hop | 96,106 | 9,992 | 9.947 |
| MetaQA 2-hop | 118,948 | 14,872 | 14,872 |
| MetaQA 3-hop | 114,196 | 14,274 | 14,274 |
| WebQSP | 2,998 | 100 | 1,639 |

Table 1: QA data statistics for each dataset according to Saxena et al. [2020]

| Dataset | Triples | Entities | Relations | Experiment-Alias |
|---|---|---|---|---|
| MetaQA-KG-Full | 135k | 43k | 9 | MetaQA_full |
| WebQSP-KG-Full | 5.7 million | 1.8 million | $\gamma$ | fbwq_full |
| MetaQA-KG-50 | $\phi$ | - | - | MetaQA_half |
| WebQSP-KG-50 | $\psi$ | - | - | fbwq_half |

Table 2: KG data statistics for each dataset. Refer[3] for more details.

Experiment-Alias is the name used for the respective datasets in experiments.
$\gamma$ = Contains all facts that are within 2-hops of any entity mentioned in the questions of WebQSP.
$\phi$ = Contains only 50% of the triples (randomly selected without replacement).
$\psi$ = Contains 50% of the edges sampled randomly from fbwq_full.

question is placed between  and  tags for all transformers except SentenceTransformer as it takes the input
sentence in its pure form. The preprocessing used in the original code has been used here. No additional preprocessing
has been performed from our end.

## 3.3 Hyper-Parameters

Hyper-parameters used to train the models aren't explicitly shared in the codebase or the paper, hence we decided to
use the default values provided in the codebase[3] to compensate the lack of time. For reproduction, a pretrained model
shared along with the data was used; ComplEx(Trouillon et al. [2016]) was used as the knowledge graph embedding
method for all the KG types, i.e., full and half of both datasets types.

| Type | MetaQA-KG-Full | | | MetaQA-KG-50 | | |
|---|---|---|---|---|---|---|
| | 1-hop | 2-hop | 3-hop | 1-hop | 2-hop | 3-hop |
| Train (t) | 350 seconds | 380 seconds | 380 seconds | 280 seconds | 330 seconds | 320 seconds |
| Validation (v) | 42 seconds | 95 seconds | 147 seconds | 47 seconds | 108 seconds | 182 seconds |
| T | 10.89 hours | 13.19 hours | 14.64 hours | 9.08 hours | 12.16 hours | 13.94 hours |

| Type | WebQSP-KG-Full | WebQSP-KG-50 |
|---|---|---|
| Train (t) | 280 seconds | 300 seconds |
| Validation (v) | 95 seconds | 105 seconds |
| T | 10.42 hours | 10.92 hours |

Table 3: Time for training/validation. Refer section:3.3 for hyper-parameters. (r=1, total_epochs=100)

For table:3, validate_every = The number of train routines before validation for a single epoch
Total runs (r) = Number of times the training has been performed for a particular task
Total train time (GPU hours) excluding early stopping, T = $(total\_epochs \times (t + v)) \times r$

For the purpose of reproducibility, hyper-parameters for training MetaQA and WebQSP QA models have been taken from section:{MetaQA[5], WebQuestionsSP[6]} of the original codebase respectively. For RoBERTa (Liu et al. [2019]), a pretrained model 'roberta-base' has been taken from HuggingFace transformers package (Wolf et al. [2020]). Other hyper-parameters are populated by default values in the codebase[3].

## 3.4 Experimental setup and code

Experiments have been performed on the NVIDIA DGX-1 server with 8xV100 GPUs, out of which 6 were used in this work. The metric used for validating the claims is Hits@1. According to Wang et al. [2019b], Hits@$k$ is the proportion of test triples ranking in the top-k results. The code for this work is open-sourced on GitHub[1]. In addition to this, we have shared couple of Docker images[7] for easy kick-starting of experiments without the hassle of setting up the environment.

Following trained models are made available in our Docker image[7], chosen on the basis of better performance in our extended study.

- TuckER KG embedding model for Meta-QA-{Full, 50}

- QA models trained using ComplEx as KG embedding model and SBERT mentioned in table:4 as question embedding model for WebQSP-KG-{Full, 50}

## 3.5 Computational requirements

This work has been performed on 6 V100-16GB GPUs connected via NVLink. NVLink reduced multi-GPU training time by 1/4. The time required for various reproductions are mentioned in table:{5,6}.

## 3.6 Extended Experiments

Apart from reproducing the results mentioned in the original paper, a couple of extended experiments have been performed to find answers to the following two questions:

1. Can recent KG embedding methods like TuckER ( Balaževi'c et al. [2019]) give higher accuracy on higher levels of hops, i.e., 3-hop scenario to be specific compared to Trouillon et al. [2016] used in the original paper?

---

[5]`https://github.com/malllabiisc/EmbedKGQA#metaqa`
[6]`https://github.com/malllabiisc/EmbedKGQA#webquestionssp`
[7]`https://github.com/jishnujayakumar/MLRC2020-EmbedKGQA#helpful-pointers`

| Transformer | Pretrained-Model |
|---|---|
| RoBERTa | roberta-base |
| XLNet | xlnet-base-cased |
| ALBERT | albert-base-v2 |
| SentenceTransformer (SBERT) | sentence-transformers/bert-base-nli-mean-tokens |
| Longformer | allenai/longformer-base-4096 |

Table 4: Pretrained models from HuggingFace transformers package (Wolf et al. [2020])

2. Can other transformer architectures like ALBERT (Lan et al. [2019]), XLNet (Yang et al. [2019]), Long-former (Beltagy et al. [2020]) and SBERT (Reimers and Gurevych [2019]) improve the results on WebQSP (Yih et al. [2016])?

Details of hyper-parameters used for these experiments are available in our GitHub repository[1]. Various transformer models used for experiment-2 are mentioned in table:4.

# 4 Results

We report results for reproducibility as well as our extended experiments. The results of reproduction have a mixed nature while the ones for our extended experiments show positive signs to support claim-1, 2. Detailed discussion about the results can be found in section:5. For all tables in this report, bold values indicate better performance.

## 4.1 Results reproducing original paper

We perform two experiments based on the two datasets introduced in section:3.2. These experiments provide vital information about the results mentioned in table:{2,5} of the original paper. The results of the two are reported in table:{5, 6} respectively. From the results of table:5 in Saxena et al. [2020] and table:6 in this report, it is evident that relation matching(RM) is an important component in multi-hop KGQA when the given KG is considerably large, i.e. {MetaQA, WebQSP} KG-Full; Definitely, WebQSP-KG-50 also shows improvement in presence of RM but the performance significantly improves when applied to KG-Full setting. The author of Saxena et al. [2020] had also expressed the same opinion in one of the virtual meetings.

## 4.2 Results beyond the original paper

We have conducted two additional experiments from our end to find an answer to claim:{1,2}. The results in table:{7,8} support claim-1 but with a caveat. On the other hand, values in table:9 improve upon the results reported by the original paper creating a new SOTA baseline. Additional experiments ingest custom hyper-parameters mentioned in our codebase in absence of the original hyper-parameters. None of these experiments include the RM module.

# 5 Discussion

The reproducibility results from table:{5,6} corroborate the claims mentioned in section:2 to some extent. The reproduced version is within $\pm 2.4$ range (positive value indicates better performance and vice-versa) except for MetaQA-KG-Full dataset's 3-hop and WebQSP-KG-Full scenario which has a significant drop of 22.5% and 18.5% respectively. The absence of RM module has been reported and discussed here[8,9,10]. For a given question, the RM module uses it's context to extract useful information from the available edges present in the KG. This information is further plugged into the answer selection module to select more relevant answers. Thus, relation matching is a vital component in multi-hop question answering, especially in KG-Full setting where more number of the edges are present w.r.t. KG-50 setting or any smaller KG w.r.t. the KG-Full setting. Results from table:{5,6} corroborates the previous statement. Moreover, MetaQA-KG-50 3-hop outperforms the original model by a margin of +0.9% without using RM which is an interesting observation. Apart from one reported anomaly, the reproduced results are pretty close to the original

---

[8] https://github.com/malllabiisc/EmbedKGQA/issues/1

[9] https://github.com/malllabiisc/EmbedKGQA/issues/51

[10] https://github.com/malllabiisc/EmbedKGQA/issues/56

| Model | RM | MetaQA-KG-Full | | | MetaQA-KG-50 | | |
|---|---|---|---|---|---|---|---|
| | | 1-hop | 2-hop | 3-hop | 1-hop | 2-hop | 3-hop |
| EmbedKGQA | ✔ | **97.5** | **98.8** | **94.8** | **83.9** | **91.8** | 70.3 |
| EmbedKGQA (Reproduced) | ✘ | 95.4 | 96.4 | 72.3 | 83.2 | 91.6 | **71.2** |
| | | | | | | | |
| $\Delta$ | - | $-2.1$ | $-2.4$ | $-22.5$ | $-0.7$ | $-0.2$ | **0.9** |

Table 5: Hits@1 results for original and reproduced experiments using MetaQA-KG-{Full, 50} datasets. $\Delta$ = (Reproduced Hits@1 without RM) - (Original Hits@1 with RM)

| Model | RM | WebQSP-KG-Full | WebQSP-KG-50 |
|---|---|---|---|
| EmbedKGQA | ✔ | **66.6** | **53.2** |
| EmbedKGQA | ✘ | 48.1 | 47.4 |
| EmbedKGQA (Reproduced) | ✘ | 54.9 | 41.3 |
| | | | |
| $\Delta_{original}$ | - | 18.5 | 5.8 |
| $\Delta$ | - | 6.8 | -6.1 |

Table 6: Hits@1 results for original and reproduced experiments using WebQSP-KG-{Full, 50} datasets. $\Delta$=(Reproduced Hits@1 without RM) - (Original Hits@1 without RM), $\Delta_{original}$ = (Original Hits@1 with RM) - (Original Hits@1 without RM).

For table:{5, 6}, KG-Embedding-Model=ComplEx. RM=Relation Matching, ✔= inclusion, ✘= exclusion. The original values for EmbedKGQA are taken from Saxena et al. [2020]. Underline indicates anomaly due to the absence of RM module.

| KG-Model | MetaQA-KG-Full | | | | | | | | |
|---|---|---|---|---|---|---|---|---|---|
| | 1-hop | | | 2-hop | | | 3-hop | | |
| | Hits@1 | Hits@5 | Hits@10 | Hits@1 | Hits@5 | Hits@10 | Hits@1 | Hits@5 | Hits@10 |
| ComplEx | 95.39 | **99.83** | 99.97 | **96.46** | **99.02** | 99.27 | 72.33 | 93.27 | 95.66 |
| TuckER | **95.51** | 99.81 | 99.97 | 93.13 | 98.7 | **99.28** | **73.81** | **93.6** | **96.09** |
| | | | | | | | | | |
| $\Delta$ | **0.12** | $-0.02$ | 0 | $-3.33$ | $-0.32$ | **0.01** | **1.48** | **0.33** | **0.43** |

Table 7: Comparison of ComplEx with TuckER based on Hits@$k$ results for MetaQA-KG-Full dataset. $k \in \{1, 5, 10\}$.

| KG-Model | MetaQA-KG-50 | | | | | | | | |
|---|---|---|---|---|---|---|---|---|---|
| | 1-hop | | | 2-hop | | | 3-hop | | |
| | Hits@1 | Hits@5 | Hits@10 | Hits@1 | Hits@5 | Hits@10 | Hits@1 | Hits@5 | Hits@10 |
| ComplEx | **83.24** | **89.83** | **91.22** | **91.63** | **97.08** | **98.04** | 71.2 | 90.77 | 93.72 |
| TuckER | 83 | 89.36 | 90.41 | 86.07 | 94.66 | 96.4 | **71.96** | **91.16** | **93.94** |
| | | | | | | | | | |
| $\Delta$ | $-0.24$ | $-0.47$ | $-0.81$ | $-5.56$ | $-2.42$ | $-1.64$ | **0.76** | **0.39** | **0.22** |

Table 8: Comparison of ComplEx with TuckER based on Hits@$k$ results for MetaQA-KG-50 dataset. $k \in \{1, 5, 10\}$.

For table:{7, 8}, $\Delta$= (TuckER Hits@$k$) - (ComplEx Hits@$k$).

| Question-Embedding-Method | WebQSP-KG-Full | | | WebQSP-KG-50 | | |
|---|---|---|---|---|---|---|
| | Hits@1 | Hits@5 | Hits@10 | Hits@1 | Hits@5 | Hits@10 |
| RoBERTa (Liu et al. [2019]) | 54.96 | 67.62 | 71.97 | 41.27 | 51.14 | 54.19 |
| XLNet (Yang et al. [2019] | 51.98 | 64.44 | 69.11 | 39.33 | 49.25 | 52.04 |
| ALBERT (Lan et al. [2019]) | 47.31 | 59.83 | 63.98 | 31.15 | 42.31 | 45.68 |
| Longformer (Beltagy et al. [2020]) | 54.9 | 66.77 | 70.47 | 41.92 | 51.98 | 54.83 |
| SBERT (Reimers and Gurevych [2019] | **55.55** | **68.98** | **72.74** | **44.65** | **53.86** | **56.13** |
| | | | | | | |
| Δ | **0.59** | **1.36** | **0.77** | **3.38** | **2.72** | **1.94** |

Table 9: Hits@$k$ results for recent transformer models by Wolf et al. [2020] used for generating question embeddings. KG-Embedding-Method=ComplEx, Δ= (SBERT_Hits@$k$ - RoBERTa_Hits@$k$), $k \in \{1, 5, 10\}$

results in case of MetaQA dataset. Default set of hyper-parameters mentioned in the original codebase(Refer section: 3.3) were used in the reproducibility study. The anomaly in WebQSP-KG-Full,i.e. 18.5% drop bolsters the importance of RM in KG-Full setting. The reproduced results for WebQSP-KG-50 are within ±7% range. The use of different hyper-parameters can be one of the possible answers to this variation. This value is significant but not w.r.t. WebQSP-KG-Full's drop of 18.5% which again strengthens the importance of RM in KG-Full setting. As mentioned in 4.1, RM is highly useful when the KG is considerably large.

From table:{7, 8}, it is clear that TuckER (Balažević et al. [2019]) performs better than ComplEx (Trouillon et al. [2016] for the 3-hop scenario for both MetaQA-KG datasets, i.e., Full and 50. Though these results strengthen claim-1, a more comprehensive set of tests may lead to a concrete conclusion. (e.g., experiments employing a broader set of hyper-parameters).

According to table:9, in all the cases, SBERT (Reimers and Gurevych [2019]) outperforms RoBERTa (Liu et al. [2019]) used in the original paper creating a new SOTA benchmark which supports claim-2.

There were some experiments which didn't work out due to the lack of time. E.g. Using RelationalTucker3 (Wang et al. [2019a]) and SimplE (Kazemi and Poole [2018]) to test claim-1. Furthermore, hyper-parameter search couldn't be done due to the same reason hence we had to pick the default ones mentioned in the codebase. All these create a room for further experiments and improvements.

## 5.1 What was easy

The paper was straightforward to understand. The open-sourced codebase helped us get kick-started.

## 5.2 What was difficult

The structure of the codebase made it difficult to navigate it. Since the code relied upon different techniques for the two datasets, the development of one function that trains different kinds of KG embeddings and another function that trains different kinds of QA models for both datasets was difficult. MetaQA uses LSTM/GRU (Hochreiter and Schmidhuber [1997] / Chung et al. [2014] while WebQSP uses RoBERTa (Liu et al. [2019]) to perform the same task of generating question embeddings. Also, training KG embeddings for MetaQA yields files in the form of NumPy (Harris et al. [2020]) files while WebQSP uses LibKGE (Broscheit et al. [2020]) for the same purpose which produces LibKGE specific KG embedding(KGE) models. Reproduction and the extensive study was a bit hard in the beginning as KGE and question embedding methodology varied for both datasets. After having a couple of virtual meetings with the author and code review, it became easier to conduct the planned experiments. The unavailability of hyper-parameters used to train each module increased the experiment cycle by multi-fold.

## 5.3 Communication with original authors

We had a couple of virtual meetings with the primary author of Saxena et al. [2020]. Though it was daunting to understand the codebase due to the reasons mentioned in section:5.2 with the help and support of the author, it became easier to navigate the codebase.

## 5.4 Future Scope

We think that there is a wide range of empirical analysis and experimentation that can be performed for multi-hop QA task, out of which we are sharing a few here:

1. KG embedding compression (Sachan [2020])
2. Using recent transformer models like Performer (Choromanski et al. [2020]), Reformer (Kitaev et al. [2020]) etc. for generating question embeddings.
3. Using low-dimensional hyperbolic KG embeddings (Chami et al. [2020]) in KG embedding module along with hyperbolic word embeddings (Dhingra et al. [2018]) for question embedding module.
4. A new approach for sentence embedding, SBERT-WK (Wang and Kuo [2020]) instead of SBERT (Reimers and Gurevych [2019]) can be tried out.

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
