# OpenReview forum: "EmbedKGQA: Improving Multi-hop Question Answering over Knowledge Graphs using Knowledge Base Embeddings"
_ML_Reproducibility_Challenge/2020 — RC2020_

### Official Review · AnonReviewer2 · 2021-03-01
**Good work, additional experiments are helpful**

**Rating:** 7
**Confidence:** 4

**Review:**

Authors replicated the work of Saxena et al. [2020] on multi-hop question answering over knowledge graphs using KB embeddings. Glad to see the core work was reproducible. The added experiments on the use of other recent KB embeddings and use of transformer architectures for question embeddings were helpful to highlight the impact of other methods on the overall framework. The absence of sufficient documentation and unavailability of hyperparameter values made the task difficult. The relation matching experiment was not possible to be replicated due to similar issues. Overall, the work appeared to be solid and would be beneficial to the community.

**Familiar With The Original Paper:**

I have not read the original paper

**Reproducibility Summary:**

Report has summary

---

### Official Review · AnonReviewer1 · 2021-03-05
**Review #1**

**Rating:** 5
**Confidence:** 4

**Review:**

The report reran the open-sourced codes from the original paper and managed to reproduce most of the results presented in the original paper. The authors also experimented with different question encoder and achieved an improvement over the original results.

The authors also cleaned up the codes, added comments, and provided command line functions.

The paper also reported hardware requirement and experiment run time.

It's glad to see that the authors try additional models beyond the original implementation. However, the authors mostly reused the default hyper-parameters, as well as the open-sourced codes from the original paper. This challenge recommends authors to either re-implement the original codes, or conduct hyper-parameter sweep. This paper failed to follow the instructions carefully.

To improve the report:
Could you please provide more details about the "relation matching" module?  It's claimed that this is the key issue that causes 22.5% drop in performance in MetaQA KG-Full 3 hop questions, but it's not discussed in the report at all. I think briefly introducing this module will make this report stand alone and help reader understand why it causes a huge drop in performance.



**Familiar With The Original Paper:**

I have read the original paper

**Reproducibility Summary:**

Report has summary

---

### Official Review · AnonReviewer3 · 2021-03-15
**Good Report! Especially appreciated the improvements from trying different KG embedding and Question embedding models.**

**Rating:** 7
**Confidence:** 4

**Review:**

In this report, the authors investigate the ACL 2020 paper by Saxena et al. on knowledge base embeddings.

Reproducibility Summary : Major findings are included in the summary.

Scope of reproducibility - Scope was clearly delineated and adhered to.

Code: Reused author repository, but made some changes to improve the code (enhance modularity) and make it easy to swap in pertained models for the question and knowledge graph embedding modules.

Communication with original authors: There was communication with the first author of the paper (virtual meetings). Efforts were made to try to address the parts of the paper that could not be reproduced, though the relation matching part of the code wasn't used which plays a part in the gap.

Hyperparameter Search: No hyper parameters sweep. Used default values. However they did tune hyper-params in their extended experiments. Things were especially difficult as little information about hyperparam values was provided in the paper and by the authors.

Ablation Study: Did do ablations over substituting the knowledge graph and question embedding modules.

Discussion on results: Descriptions on what was easy and difficult were provided. The scope of the reproduction was to reproduce results in the original paper which they were able to get close on one dataset (MetaQA) but remain far apart on the second (WebQSP) dataset, 11.7 percent absolute despite correspondence with the authors (although the reason is lack of relation matching)

The authors did extra work to test different knowledge graph embedding models and different transformer models for question embeddings. They found improvements by using TuckER and Sentence Bert respectively.

Overall organization and clarity: Paper is clear and organized and easy to read. It feels complete to me as well, showing what can be reproduced and what cannot - although more discussions on where the divergence for QebQSP could've been provided and the experiments without the relation matching make it unclear how close reproduction was performed for WebQSP.

**Familiar With The Original Paper:**

I have not read the original paper

**Reproducibility Summary:**

Report has summary

---

### Decision · Program_Chairs · 2021-03-31

**Decision:**

Accept

**Comment:**

Selected for ReScience-C Journal Publication.
This paper thoroughly reproduces the results from the original paper, and includes some results which were not able to be reproduced (a valuable finding, which will likely be examined by others looking to build upon the original work). In addition, the authors extend the original work and evaluate with different models, getting a new SOTA baseline.